# Speed Eating Is Associated with Poor Mental Health Among Adolescents and Young Adults: A Cross-Sectional Study

**DOI:** 10.3390/nu17172822

**Published:** 2025-08-29

**Authors:** Yuko Fujita, Tomohiro Takeshima

**Affiliations:** 1Development and Nurturing Dentistry, Graduate School of Biomedical Sciences, Nagasaki University, Nagasaki 852-8588, Japan; 2Takeshima Dental Office, Okinawa 904-2143, Japan; tomohiro1207045@gmail.com

**Keywords:** swallowing threshold, speed eating, mental health, adolescents, young adults

## Abstract

Background: This study aimed to determine whether mental health status contributes to speed eating in adolescents and young adults. Methods: This study enrolled 106 subjects (53 males and 53 females), ranging in age from 12 to 24 years. After a self-administered lifestyle questionnaire and the 12-item General Health Questionnaire (GHQ-12) were administered, a swallowing threshold test was performed. The swallowing threshold was determined based on the concentration of dissolved glucose obtained from the gummy jellies. Low swallowing threshold was characterized by glucose levels falling within the bottom 20th percentile. GHQ-12 was categorized into poor (score 4–12) and normal (score 0–3). Following the univariate analysis, a multivariate binary logistic regression analysis was conducted to determine the factors linked to a low swallowing threshold. Results: Binomial logistic regression analysis revealed that the factors associated with a low swallowing threshold included poor mental health (odds ratio [OR] = 8.47, *p* = 0.007, confidence interval [CI] = 2.437–32.934) and no physical activity (OR = 5.604, *p* = 0.008, CI = 1.562–22.675). Conclusions: Speed eating is closely associated with risk behaviors for poor mental health in adolescents and young adults.

## 1. Introduction

Various types of abnormal eating behaviors, including imbalances in food intake, meal content, and abnormal eating habits, are known [1,2,3,4,5]. Recent reports have shown that eating disorders are associated with poor mental health, which may lead to under- or over-eating [6,7], and another study reported that individuals with high levels of emotional eating and uncontrolled eating behaviors are at a higher risk of poor mental health [8]. Thus, changes in mental health during adolescence may contribute to or hinder healthy eating habits, and may even perpetuate poor eating habits and abnormal eating behaviors in the future.

Among the abnormal eating behaviors, speed eating has recently begun to attract particular attention. Several studies have reported the relationship between speed eating and overall health. Individuals who eat speedy are more likely to face a higher likelihood of develop diabetes and obesity [9,10]. Moreover, speed eating may lower the thermic effect of food, possibly because a reduction in chewing frequency diminishes sympathetic nervous system activity in young women [11].

Recent studies have reported that bite force and chewing ability increase with age, peaking around the age of 30 years, but the growth patterns of the number of chews and chewing time before swallowing fluctuate within a small range after the age of 5 years, with no significant regression even after the age of 30 years [12,13]. It has also been reported that a stable swallowing threshold might be acquired at a relatively early age [12]. Therefore, if speed eating is associated with poor mental health in adolescents and young adults, it may persist throughout life. However, few studies have focused on chewing movements, including speed eating, and mental health in adolescents and young adults. Based on these studies, we hypothesized that a low swallowing threshold, including the number of chews, chewing time, and chewing rate before swallowing, are associated with poor mental health in adolescents and young adults.

Subjective methods are commonly used to evaluate eating speed in epidemiological studies [9,10,14,15]. However, there are problems with reliability and reproducibility when self-administered questionnaires are used to evaluate the swallowing thresholds. Previous studies have been reported that each person has a certain swallowing threshold [16], and the number of chews and chewing time required to swallow the food are roughly constant depending on each person’s habits [17]. Therefore, we decided to use a method that would be as objective as possible. In this study, the swallowing threshold was defined as the moment when various sensory nerves determined that the chewed gummy jellies had physical properties suitable for swallowing. We identified the swallowing threshold by measuring the concentration of dissolved glucose from the gummy jellies at the moment participants expressed their desire to swallow them. The total surface area of the chewed gummy jelly pieces is converted into the amount of glucose dissolved in the gummy jelly. To clarify the cause of a low swallowing threshold, i.e., speed eating, number of chews, chewing time, and chewing rate before swallowing were also included in the evaluation items. However, the “swallowing threshold” by this method is an exploratory indicator, and not a standardized or universally accepted measure of chewing behavior. This study aimed to clarify whether mental health status contributes to reduced chewing before swallowing in adolescents and young adults aged 12–24 years.

## 2. Materials and Methods

### 2.1. Participants

This study was approved by the Human Investigation Committee at Kyushu Dental University (Kitakyushu, Fukuoka, Japan; approval number 22–37). Participants visited Kyushu Dental University Hospital for the first time and were selected after an initial examination from May 2023 to March 2024. Informed consent was obtained from all participants before joining the study. For participants aged 12–24 years, parental consent was obtained consent to take part.

A priori power analysis was conducted using G*Power software (version 3.1.9.4 for Windows; accessible from the Heinrich-Heine-Universität Düsseldorf website) to determine the sample size. This analysis considered an odds ratio (OR) of 4, a 0.4 incidence of low swallowing threshold in the group with normal mental health, a power of 0.90, and a type I error probability of 0.05 for the null hypothesis in the logistic regression model. The results indicate that 97 participants rejected the null hypothesis. Participants eligible for the study were aged between 12 and 24 years, in good physical condition, and without any issues with language comprehension, existing dental diseases, or complications. Individuals with systemic conditions causing dysfunction of the tongue or perioral muscles, noticeable irregularities in facial symmetry that could influence measurements, soft tissue irregularities, temporomandibular joint issues, abnormal tooth shapes, negative overjet, overbite, or those currently receiving orthodontic treatment were excluded from the study. A total of 106 participants were recruited for this study, as determined by power analysis.

### 2.2. Questionnaire

The 12-item Japanese version of the General Health Questionnaire (GHQ-12) was used to assess mental health status (Table 1) [18,19]. Participants indicated whether they had recently encountered a specific symptom of mental distress using a bimodal scale (0–0–1–1, where the two least symptomatic responses were scored as 0 and the two most symptomatic responses were scored as 1). For convenience, participants whose mental health was predicted to worsen based on the assessment of risk behaviors were defined as poor mental health (scores 4–12), while the remaining participants had normal mental health (scores 0–3) [20].

The survey requested details on the following topics: sex, dietary patterns, exercise routines, and sleep habits (Table 2).

### 2.3. Anthropometry and Dental Examination

Height and body weight were measured in the consultation rooms. A portable digital stadiometer (AD-653; A&D, Tokyo, Japan) was used to measure height with a precision of ±0.1 cm, while weight was recorded with a precision of 0.1 kg. BMI was calculated based on the respective height and weight measurements according to age. Degree of overweight or obesity was determined according to the World Health Organization criteria.

During the intraoral examination, Decayed, Missing, and Filled Tooth (DMFT) indices were recorded for all participants.

### 2.4. Maximum Occlusal Force

Peak occlusal force was assessed using a portable occlusal force meter (GM10; Nagano Keiki, Tokyo, Japan), which incorporated a strain gauge positioned at the center of the bite component encased in a plastic tube. The participants were seated comfortably during the evaluation. They were instructed to place the bite element on their upper first molars and apply maximum voluntary muscle force for approximately three seconds. The element was equipped with a pressure gauge to record the occlusal force in kilonewtons (kN), and the results were digitally displayed. The highest value recorded on either the left or the right side was considered the maximum occlusal force and was used for further analysis [21].

### 2.5. Masticatory Performance

The determination of masticatory efficiency involved measuring the glucose concentration released from cylindrical gummy jelly (GLUCOLUMN, GC Corporation, Tokyo, Japan). Participants chewed gummy jelly for 20 s using their preferred side for chewing. Afterward, they rinsed their mouths with 10 mL of distilled water and expelled the gummy jelly, water, and saliva into a cup with a filter. Glucose concentration (mg/dL) in the filtered solution was measured using a specialized device (GLUCO SENSOR GS-II, GC Corporation) [22].

### 2.6. Swallowing Threshold

After assessing the masticatory performance, the swallowing threshold was measured using the same jelly (GLUCOLUMN, GC Corporation). Variables associated with swallowing threshold, such as the number of chews (N), chewing time (s), chewing rate (s/N), and glucose level in the filtrate (mg/dL), were analyzed. Participants were asked to chew the gummy jelly at their own pace until they felt the need to swallow, at which point they signaled the examiner by raising their hand. The examiner counted the number of chews visually. The time from the beginning of chewing to the moment the participants raised their hand was recorded using a stopwatch. The subsequent steps mirrored those used to assess the masticatory performance. Referring to a previous study that outlined sarcopenia criteria, we identified a developmental disorder of the swallowing threshold when the glucose level fell within the lowest 20th percentile [12].

### 2.7. Reliability of Measurements

All assessments were conducted by a single examiner, with a 30-min interval between each test. The data’s consistency was assessed for random error through intraobserver reliability, using intraclass correlation coefficients (ICC; 0.8 ≤ ICC ≤ 1.0 indicates high reliability) [23]. The ICCs for all measurements varied between 0.8 and 0.95, indicating high reliability.

### 2.8. Data Analyses

The Shapiro–Wilk test was employed to assess whether the data followed a normal distribution. Each group’s measurements are displayed as the mean ± standard deviation. A two-tailed *t*-test was used to compare mean values between the two groups. The chi-square test was used to examine differences in categorical variables between participants with normal and low swallowing thresholds. Residual analysis was conducted when either the chi-squared test or Fisher’s exact test was applied to evaluate a significant relationship between the two variables. Pearson’s correlation coefficient was used to explore the relationship between age, anthropometry, DMFT index, GHQ-12 score, and oral function. To identify the factors linked to a low swallowing threshold, a binary logistic regression analysis was conducted using the forward selection (conditional) approach. Variables that showed significance in the univariate analysis were incorporated. Categorical variables were coded appropriately before their inclusion in the model. Adjusted odds ratios (ORs) and their 95% confidence intervals (CIs) were determined for the group with low swallowing threshold. All analyses were performed using the SPSS software (version 23.0 for Windows, IBM Japan, Tokyo, Japan), with statistical significance set at *p* < 0.05.

## 3. Results

Table 2 presents the findings from the cross-tabulation analysis of the swallowing threshold in relation to sex, obesity level, mental health, and lifestyle habits using either the chi-square test or Fisher’s exact test. Participants with a low swallowing threshold were more likely to have poor mental status, eating once or more a day between meals, no physical activity, and poor sleep quality (*p* = 0.002, *p* = 0.002, *p* = 0.013, and *p* = 0.021, respectively).

Table 3 compares age, height, body weight, BMI, DMFT index, and oral function, all categorized by sex, between the groups with normal and low swallowing thresholds. The GHQ-12 score was significantly higher in the low swallowing threshold group than in the normal swallowing threshold group in males and females (both *p* < 0.05), whereas the maximum occlusal force, masticatory performance, number of chews, and chewing time were significantly lower in the low swallowing threshold group in males and females (all *p* < 0.05).

Table 4 compares age, height, body weight, BMI, DMFT index, and oral function between the groups with normal and low swallowing thresholds. The GHQ-12 score was significantly higher in the low swallowing threshold group than in the normal swallowing threshold group (*p* < 0.05), whereas the maximum occlusal force, masticatory performance, number of chews, and chewing time were significantly lower in the low swallowing threshold group (all *p* < 0.05).

Table 5 summarizes the results of the Pearson’s bivariate correlation analyses. Higher GHQ-12 scores (indicating worse mental health) were associated with fewer number of chews, shorter chewing time, and lower swallowing threshold (*r* = −0.372, *p* < 0.01, *r* = −0.379, *p* < 0.01, and *r* = −0.371, *p* < 0.01, respectively). Number of chews was significantly positively associated with chewing time (*r* = 0.800, *p* < 0.01). The swallowing threshold was significantly positively associated with chewing time (*r* = 0.602, *p* < 0.01), the number of chews (*r* = 0.584, *p* < 0.01), and masticatory performance (*r* = 0.527, *p* < 0.01).

Table 6 presents the elements linked to a low swallowing threshold as identified through a binomial logistic regression analysis. The analysis identified poor mental health (OR = 8.470, *p* = 0.007, CI = 2.437–32.934) and absence of physical activity (OR = 5.604, *p* < 0.008, 95% CI = 1.562–22.675) as significant factors contributing to a low swallowing threshold.

## 4. Discussion

Logistic regression analysis revealed that poor mental health was significantly associated with low swallowing threshold. Consistent with our findings, a recent meta-analysis also suggest that poor mental health is associated with unhealthy eating behaviors in children [24]. The bivariate analysis revealed a significant negative correlation between the GHQ-12 score and both the number of chews and the chewing time. These results suggest that the more behaviors participants engaged in that are linked to poor mental health, the more likely they were to swallow gummy jellies without chewing them properly. However, the chewing rates were approximately the same in both the low and normal swallowing threshold groups. These findings suggest that the lower swallowing threshold was due to fewer chews and shorter chewing times before swallowing, rather than a faster chewing speed (sec per cycle). In participants with high GHQ-12 scores, poor mental health may have altered eating behaviors through abnormalities at the neural circuit level including mesolimbic (reward) circuit and stress/motivation circuits [25]. It is also possible that regular speed eating contributes to poor mental health. In addition, recent studies have reported that eating speed has associated with insufficient food parenting practices [26], higher number of siblings [27], snacking before sleep or supper [28], and eating with fingers or spoons rather than chopsticks [29]. These findings suggest that speed eating is related to complex social, environmental and cultural factors in addition to psychological factors.

The GHQ-12 is a widely used tool to screen for general psychological distress. It has been employed across various populations and countries to assess its reliability and validity, as well as to investigate the characteristics of psychological distress [30,31,32]. Several studies have included adolescent males and females as subjects [32,33,34]. Regarding the relationship between GHQ-12 and eating behavior in young people, a few studies have reported that Eating Attitude Test-26 (EAT-26) scores indicating disordered eating attitudes are associated with GHQ-12 scores among university students and young adults [35,36]. There may be a strong general perception that speed eating is not as serious a problem as eating disorders, but in order to promote mental and physical health and prevent various diseases, it is necessary to improve the speed eating habit of patients as early as possible.

Logistic regression analysis showed that risk factors for a low swallowing threshold included not only poor mental health but no physical activity. It has been reported that lack of physical activity is a known determinant of metabolic syndrome [37]. In addition, a higher risk for metabolic syndrome was significantly associated with speed eating and physical inactivity in university students [38]. These results suggest that lack of physical activity and low swallowing threshold may be indirectly related to metabolic syndrome.

Several studies have reported that speed eating is linked to increased body fat, higher BMI, and a greater likelihood of being overweight or obese in adults [39,40,41] and university students [42], indicating that speed eating contributes to obesity. Another study indicated that alterations in the pace of eating can influence obesity, BMI, and waist circumference [43]. Our findings also showed that higher BMI was associated with fewer number of chews and shorter chewing time. In this study, we used same type of gummy jelly as the test food to assess 20 s masticatory performance and swallowing threshold. As the results, the 20 s masticatory performance and maximum occlusal force in the low swallowing threshold group were significantly lower than in the normal swallowing threshold group. These results suggest that participants with low swallowing thresholds have weaker occlusal force, fewer chewing cycles, or both. Therefore, we believe that our swallowing threshold assessment is useful and reliable. However, it should be noted that the “swallowing threshold” by our method is an exploratory indicator, and not a standardized or universally accepted measure of chewing behavior. It has been reported that obese children have lower taste sensitivity than non-obese children, indicating that they require a greater amount of food to taste it [44]. We need to consider that chewing speed cannot be determined by a single stimulus alone, as some participants may not have been accustomed to chewing gummy jelly, while others may have been.

Univariate analysis revealed that poor sleep status and snacking habits are associated with a low swallowing threshold. Because all participants who reported poor sleep quality gave the same answers to the GHQ-12 questionnaire, there may be an association with GHQ-12 scores. Therefore, there may be an indirect relationship between poor sleep quality and a low swallowing threshold. The strength of the association between each factor varies depending on the age and sex of the subjects, and may vary depending on the situation, even within the same individual.

Our study has certain limitations. First, it employed a cross-sectional approach, which prevented us from establishing causal links between relevant elements. Longitudinal research is necessary to explore the relative impact of the factors associated with speed eating. Second, The GHQ questionnaire is quite broad and only shows current risk behaviors. It suffers from a lack of diagnostic specificity, additional tests are needed to adequately support the mental health outcomes of the participants. Participants’ current risk behaviors may be related to social, economic, relationship, and self-image problems. All subjects were students and there were no subjects of other social and economic levels, it was not possible to fully extrapolate the results. Therefore, the recruitment process and potential biases in sample composition, and limited socioeconomic diversity and age composition of the participants in this study may limit external validity. In the future, we need to analyze factors related to speed eating in subjects with various socioeconomic situations.

## 5. Conclusions

We found that reduced chewing before swallowing is closely associated with risk behaviors for poor mental health in adolescents and young adults. Additionally, sleep quality and dietary patterns may also be associated with poor chewing behavior. Screening for eating speed should be included in mental health evaluations in schools and adolescent clinics.

## Figures and Tables

**Table 1 nutrients-17-02822-t001:** GHQ-12 items.

GHQ-12 Items
Have you…
been able to concentrate on what ever you are doing?
2. lost much sleep over worry?
3. felt that you were playing a useful part in things?
4. felt capable of making decisions about things?
5. felt constantly under strain?
6. felt that you could not overcome your difficulties?
7. been able to enjoy your normal day-to-day activities?
8. been able to face up to your problems?
9. been feeling unhappy and depressed?
10. been losing confidence in yourself?
11. been thinking of yourself as a worthless person?
12. been feeling reasonably happy, all things considered?

**Table 2 nutrients-17-02822-t002:** Cross-tabulation analysis between the swallowing threshold and sex and health-related variables.

Participants (*n* = 106)	Normal (%) (*n* = 83)	Low (%) (*n* = 23)	*χ* ^2^	*p*-Value
Sex				
Female	41 (49.4)	12 (52.2)		
Male	42 (50.6)	11 (47.8)		
			―	^†^ 1.000
Degree of obesity				
Normal	42 (50.6)	8 (34.8)		
Underweight/severely underweight	18 (21.7)	5 (21.7)		
Overweight/obese	23 (27.7)	10 (43.5)		
			2.394	^‡^ 0.302
Mental health (GHQ-12 score)				
Normal (Score 0–3)	75 (90.4)	14 (60.9)		
Poor (Score 4–12)	8 (9.6)	9 (39.1)		
			―	^†^ 0.002
Skipping breakfast				
Less than two times a week	45 (54.2)	13 (56.5)		
Two times or more a week	38 (45.8)	10 (43.5)		
			―	^†^ 1.000
Snacks				
Less than once a day	52 (62.7)	6 (26.1)		
Once or more a day	31 (37.3)	17 (73.9)		
			―	^†^ 0.002
Sweaty exercise for 30 min or more per day				
Twice a week or more	35 (42.2)	5 (21.7)		
Less than twice a week	35 (42.2)	8 (34.8)		
Not currently	13 (15.7)	10 (43.5)		
			8.656	^‡^ 0.013
Self-assessed sleep quality				
Good	64 (77.1)	12 (52.2)		
Poor	19 (22.9)	11 (47.8)		
			―	^†^ 0.021

^†^ Fisher’s exact test, ^‡^ Chi-square test.

**Table 3 nutrients-17-02822-t003:** Comparison of age, height, body weight, BMI, DMFT index, and oral function, categorized by sex, between the groups with normal and low swallowing thresholds.

	Normal (*n* = 83)		Low (*n* = 23)	
	Male (*n* = 41)	Female (*n* = 42)	Male (*n* = 12)	Female (*n* = 11)
Age (years)	18.34 ± 3.73	17.79 ± 3.58	17.83 ± 3.49	19.18 ± 3.54
Height (m)	1.69 ± 0.07	1.55 ± 0.05	1.71 ± 0.06	1.58 ± 0.04
Body weight (kg)	64.16 ± 8.14	47.83 ± 5.87	65.83 ± 12.41	52.91 ± 8.35
BMI	23.77 ± 3.71	21.38 ± 3.12	24.17 ± 5.29	22.11 ± 3.38
GHQ-12 score	1.05 ± 1.70	0.93 ± 1.96	2.83 ± 2.48 *	3.73 ± 3.26 ^†^
DMFT index	4.32 ± 3.86	3.36 ± 4.48	3.50 ± 2.07	4.09 ± 3.11
Maximum occlusal force (kN)	0.49 ± 0.08	0.45 ± 0.08	0.45 ± 0.06	0.39 ± 0.09 ^†^
Masticatory performance (mg/dL)	178.80 ± 41.87	175.86 ± 36.00	145.50 ± 44.11 *	138.45 ± 38.36 ^†^
Number of chews (N)	20.76 ± 5.78	23.00 ± 6.19	12.58 ± 3.60 *	16.36 ± 7.51 ^†^
Chewing time (s)	15.85 ± 4.44	18.79 ± 4.99	10.13 ± 3.08 *	12.07 ± 4.82 ^†^
Chewing rate (s/N)	0.78 ± 0.16	0.83 ± 0.15	0.75 ± 0.21	0.76 ± 0.14
Swallowing threshold (mg/dL)	152.61 ± 31.01	156.55 ± 32.51	80.92 ± 17.11 *	82.73 ± 19.92 ^†^

Data are expressed as mean ± standard deviation. BMI, body mass index; DMFT, decayed, missing, and filled teeth. Differences between normal and low swallowing threshold groups were assessed by a two-tailed *t*-test. * *p* < 0.05 vs. normal group in males. ^†^
*p* < 0.05 vs. normal group in females.

**Table 4 nutrients-17-02822-t004:** Comparison of the age and other measurement parameters between the normal and low swallowing threshold groups of all participants.

	Normal (*n* = 83)	Low (*n* = 23)
Age (years)	18.06 ± 3.64	18.48 ± 3.50
Height (m)	1.62 ± 0.09	1.65 ± 0.09
Body weight (kg)	55.90 ± 10.83	59.65 ± 12.34
BMI	22.56 ± 3.61	23.19 ± 4.51
GHQ-12 score	0.99 ± 1.82	3.26 ± 2.85 *
DMFT index	3.83 ± 4.19	3.78 ± 2.58
Maximum occlusal force (kN)	0.47 ± 0.08	0.42 ± 0.08 *
Masticatory performance (mg/dL)	177.31 ± 38.80	142.13 ± 40.68 *
Number of chews (N)	21.89 ± 6.06	14.39 ± 5.99 *
Chewing time (Sec)	17.34 ± 4.92	11.06 ± 4.04 *
Chewing rate (Sec/N)	0.80 ± 0.15	0.76 ± 0.17
Swallowing threshold (mg/dL)	154.60 ± 31.65	81.78 ± 18.10 *

Data are expressed as mean ± standard deviation. DMFT, decayed, missing, and filled teeth; Swallowing threshold, glucose concentration on first swallow. Differences between normal and low groups were assessed by a two-tailed *t*-test. * *p* < 0.05 vs. normal group.

**Table 5 nutrients-17-02822-t005:** Peason’s correlation coefficients between the swallowing threshold and the age and other parameters.

	Age	Height	Body Weight	BMI	GHQ-12	DMFT Index	Maximum Occlusal Force	MP	Number of Chews	Chewing Time	Chewing Rate
Number of chews	0.083	−0.188	−0.249 *	−0.233 *	−0.372 **	0.138	−0.003	0.030	―	0.800 **	−0.203 *
Chewing time	−0.009	−0.254 *	−0.302 **	−0.280 **	−0.379 **	0.200 *	−0.158	−0.116	0.800 **	―	0.349 **
Chewing rate	−0.120	−0.138	−0.107	−0.109	−0.018	0.108	−0.220 *	−0.245 *	−0.203 *	0.349 **	―
Swallowing threshold	−0.004	−0.165	−0.245 *	−0.165	−0.371 **	0.028	0.272 **	0.527 **	0.584 **	0.602 **	0.100

BMI, body mass index; GHQ, General Health Questionnaire; DMFT, decayed, missing, and filled teeth; MP, Masticatory performance. * *p* < 0.05, ** *p* < 0.01.

**Table 6 nutrients-17-02822-t006:** Factors related to the low swallowing threshold based on binary logistic regression analysis.

Independent Variables	Category	Adjusted Odds Ratio (95% CI)	*p*-Value
Mental health (GHQ-12 score)	Normal (Score 0–3)	1	―
	Poor (Score 4–12)	8.470 (2.437–32.934)	0.007
Physical activity	Twice a week or more	1	―
	Less than twice a week	0.943 (0.233–3.768)	0.933
	Not currently	5.604 (1.562–22.675)	0.008

GHQ, General Health Questionnaire; CI, confidence interval.

## Data Availability

Data are contained within the article.

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
