# Peer review of "Speed Eating Is Associated with Poor Mental Health Among Adolescents and Young Adults: A Cross-Sectional Study"

_nutrients, 2025, doi:10.3390/nu17172822_

Round 1
Reviewer 1 Report
Comments and Suggestions for Authors
Manuscript: "Speed eating is associated with poor mental health among adolescents and young adults: A Cross-Sectional Study"
# Global evaluation
This is an original and well-executed cross-sectional study investigating the relationship between speed eating and mental health in adolescents and young adults. The use of objective physiological markers, such as swallowing threshold and glucose release, lends the study methodological rigour and distinguishes it from many prior investigations that rely on self-reported data.
The study is strengthened by a clear research question and a robust rationale. The objective assessment methods provide more reliable insights into eating behaviour, and the data analysis is thorough and appropriate for the study design. The authors demonstrate awareness of the study’s limitations, and the topic is clearly relevant to adolescent health and preventive care strategies.
However, there are areas that require improvement. The manuscript contains minor language and grammar issues that affect clarity. There is sometimes conceptual confusion or a lack of precise definitions, particularly around the operational meaning of 'speed eating'. Additionally, the language sometimes implies causality, which is inappropriate for a cross-sectional design. Finally, there is slight redundancy in phrasing and some terminology is used inconsistently throughout the text. Addressing these issues would enhance the clarity and overall quality of the manuscript.
# Conceptual Issues
-The study uses "low swallowing threshold" as a proxy for speed eating. While this is innovative, it’s not a standard or widely understood equivalence. You should clarify more clearly how this measure correlates with real-world eating speed.
- The manuscript occasionally implies causal relationships (e.g., poor mental health leads to speed eating), even though the cross-sectional design only supports associations. It would strengthen the scientific rigor to rephrase these claims to emphasize correlation, not causality.
-The results show no significant difference in chewing rate, yet there is a large difference in chewing time and number of chews. Be careful in how you interpret "speed": is it the rate of chewing or the total duration before swallowing?
#Mistakes by lines
-Lines 68–70.
- "they gave their informed consent was obtained..." change to: "Informed consent was obtained from all participants before joining the study."
- "consent to take part consent to take part." Delete the redundancy.
-Lines 20–21. "p = 7.95% CI" correct to: "p = 0.007, 95% CI = ..."
-Line 226. "spped eating habit" change to: "speed eating habit"
- Conclusion, last sentence. "Additionally, less chewing before swallowing is indirectly linked to sleeping habits..." change to: "Additionally, poor chewing behaviour may be influenced by sleep quality and eating patterns."
-Results. “GHQ-12 score was significantly negatively correlated,” change to the simplified text: “Higher GHQ-12 scores (indicating worse mental health) were associated with fewer chews and shorter chewing time.”
# Recommendations
-The interpretation that poor mental health causes speed eating needs to be softened.
-Acknowledge that reverse causation is possible; e.g., poor dietary habits might worsen mood.
- Expand briefly on how speed eating might physiologically relate to emotional distress, e.g., via sympathetic nervous system involvement or impulsivity.
- Consider briefly mentioning cultural factors, especially since eating behaviours vary widely across populations.
# Suggestions for Improvement
- Be explicit about how "speed eating" is operationally defined and how it relates to the chewing/swallowing metrics.
-Use consistent terms for chewing measures (e.g., "chewing time", "number of chews", "chewing rate") across the entire manuscript.
-Avoid redundancy and unclear constructions. A thorough proofreading or a professional language edit would benefit the overall readability.
-The paper references a good number of studies but could benefit from further integration with recent meta-analyses on adolescent mental health and dietary behavior.
- Conclusions are valid but could be more actionable. For instance, suggest screening for eating speed during mental health evaluations in schools or youth clinics.
Author Response
The comments have been helpful for us to revise our manuscript. We have therefore thoroughly revised the manuscript on the basis of your suggestions. The parts of the manuscript that have been significantly revised are shown in red font.
-The study uses "low swallowing threshold" as a proxy for speed eating. While this is innovative, it’s not a standard or widely understood equivalence. You should clarify more clearly how this measure correlates with real-world eating speed.
In accordance with your suggestions, we added the description of the definition and use of the "swallowing threshold" in Introduction section.
- The manuscript occasionally implies causal relationships (e.g., poor mental health leads to speed eating), even though the cross-sectional design only supports associations. It would strengthen the scientific rigor to rephrase these claims to emphasize correlation, not causality.
In accordance with your suggestions, we have changed descriptions that suggest causality to expressions that show correlation.
-The results show no significant difference in chewing rate, yet there is a large difference in chewing time and number of chews. Be careful in how you interpret "speed": is it the rate of chewing or the total duration before swallowing?
In accordance with your suggestions, we added "(seconds per cycle)" after the chewing speed.
-Lines 68–70.
"they gave their informed consent was obtained..." change to: "Informed consent was obtained from all participants before joining the study."
"consent to take part consent to take part." Delete the redundancy.
In accordance with your suggestions, we corrected the sentences.
-Lines 20–21. "p = 7.95% CI" correct to: "p = 0.007, 95% CI = ..."
In accordance with your suggestions, we corrected the values.
-Line 226. "spped eating habit" change to: "speed eating habit"
In accordance with your suggestions, we corrected the words.
- Conclusion, last sentence. "Additionally, less chewing before swallowing is indirectly linked to sleeping habits..." change to: "Additionally, poor chewing behaviour may be influenced by sleep quality and eating patterns."
In accordance with your suggestions, we corrected the sentence.
-Results. “GHQ-12 score was significantly negatively correlated,” change to the simplified text: “Higher GHQ-12 scores (indicating worse mental health) were associated with fewer chews and shorter chewing time.”
In accordance with your suggestions, we corrected the sentence.
-The interpretation that poor mental health causes speed eating needs to be softened.
We have revised the statement regarding this interpretation.
-Acknowledge that reverse causation is possible; e.g., poor dietary habits might worsen mood.
In accordance with your suggestions, we added the description of the reverse causation is possible.
- Expand briefly on how speed eating might physiologically relate to emotional distress, e.g., via sympathetic nervous system involvement or impulsivity.
In accordance with your suggestions, we added the description of the sympathetic nervous system involvement or impulsivity.
- Consider briefly mentioning cultural factors, especially since eating behaviours vary widely across populations.
In accordance with your suggestions, we added the description of the cultural factors.
- Be explicit about how "speed eating" is operationally defined and how it relates to the chewing/swallowing metrics.
We added the description of the speed eating and swallowing threshold in Introduction section.
-Use consistent terms for chewing measures (e.g., "chewing time", "number of chews", "chewing rate") across the entire manuscript.
In accordance with your suggestions, we corrected the words.
-Avoid redundancy and unclear constructions. A thorough proofreading or a professional language edit would benefit the overall readability.
In accordance with your suggestions, we corrected the construction of the text.
-The paper references a good number of studies but could benefit from further integration with recent meta-analyses on adolescent mental health and dietary behavior.
In accordance with your suggestions, we added the meta-analytic literature on adolescent mental health and dietary behavior in Discussion section.
- Conclusions are valid but could be more actionable. For instance, suggest screening for eating speed during mental health evaluations in schools or youth clinics.
In accordance with your suggestions, we added the sentences “Screening for eating speed should be included in mental health evaluations in schools and adolescent clinics.”

Reviewer 2 Report
Comments and Suggestions for Authors
The article explores an important but complex topic—the relationship between chewing behavior and mental health in adolescents and young adults—yet several key conceptual and methodological weaknesses undermine the strength and clarity of its findings. First and foremost, the definition of "poor mental health" is absent throughout the text. The term is used inconsistently, at times being replaced with "mental stress" or "mental health deterioration," and is loosely associated with a broad and heterogeneous range of issues, from clinical depression and anxiety to transient feelings like unhappiness or low confidence. These are not equivalent, nor are they interchangeable, and treating them as such raises serious concerns about the conceptual rigor of the study. Furthermore, the study relies exclusively on the GHQ-12 questionnaire to assess mental health status. While GHQ-12 is widely used as a screening tool for general psychological distress, it lacks diagnostic specificity and is insufficient on its own to substantiate claims about mental health outcomes. The categorization into "normal" and "poor" mental health based on this instrument is oversimplified and should be interpreted with caution.
Another major issue lies in the definition and use of the "swallowing threshold," a central variable in the study that is never clearly or consistently defined. It appears to refer to the point at which participants choose to swallow a test gummy jelly, assessed by glucose concentration in the filtrate, but the physiological or clinical meaning of this threshold remains ambiguous. Is it a proxy for chewing efficiency? Impulse control? Sensory response? Without a robust conceptual framework, the link to mental health remains speculative.
The discussion section is particularly weak and fails to offer a clear interpretation of the findings. There is no serious engagement with alternative explanations, nor any attempt to situate the results within existing theoretical or clinical models. Claims such as speed eating being a consequence of poor mental health are made without addressing the possibility of reverse causation or shared confounding factors such as stress, socioeconomic status, or dietary habits.
In its current form, the manuscript lacks the theoretical depth, terminological precision, and interpretive caution expected of research dealing with adolescent mental health. Clarifying key constructs, justifying the choice of instruments, and offering a more nuanced discussion of the findings are essential steps before the manuscript can be considered for publication.
Author Response
The comments have been helpful for us to revise our manuscript. We have therefore thoroughly revised the manuscript on the basis of your suggestions. The parts of the manuscript that have been significantly revised are shown in red font.
The article explores an important but complex topic—the relationship between chewing behavior and mental health in adolescents and young adults—yet several key conceptual and methodological weaknesses undermine the strength and clarity of its findings. First and foremost, the definition of "poor mental health" is absent throughout the text. The term is used inconsistently, at times being replaced with "mental stress" or "mental health deterioration," and is loosely associated with a broad and heterogeneous range of issues, from clinical depression and anxiety to transient feelings like unhappiness or low confidence. These are not equivalent, nor are they interchangeable, and treating them as such raises serious concerns about the conceptual rigor of the study. Furthermore, the study relies exclusively on the GHQ-12 questionnaire to assess mental health status. While GHQ-12 is widely used as a screening tool for general psychological distress, it lacks diagnostic specificity and is insufficient on its own to substantiate claims about mental health outcomes. The categorization into "normal" and "poor" mental health based on this instrument is oversimplified and should be interpreted with caution.
In accordance with your suggestions, we added the description of the Interpretation of the mental health categorization in this study in Methods section. Additionally, we have changed our interpretation of the GHG-12 results.
Another major issue lies in the definition and use of the "swallowing threshold," a central variable in the study that is never clearly or consistently defined. It appears to refer to the point at which participants choose to swallow a test gummy jelly, assessed by glucose concentration in the filtrate, but the physiological or clinical meaning of this threshold remains ambiguous. Is it a proxy for chewing efficiency? Impulse control? Sensory response? Without a robust conceptual framework, the link to mental health remains speculative.
In accordance with your suggestions, we added the description of the definition of the "swallowing threshold" in Introduction section.
The discussion section is particularly weak and fails to offer a clear interpretation of the findings. There is no serious engagement with alternative explanations, nor any attempt to situate the results within existing theoretical or clinical models. Claims such as speed eating being a consequence of poor mental health are made without addressing the possibility of reverse causation or shared confounding factors such as stress, socioeconomic status, or dietary habits.
In accordance with your suggestions, we added the descriptions of the possibility of reverse causation and relationships between speed eating and stress, socioeconomic status, or dietary habits in Discussion section.
In its current form, the manuscript lacks the theoretical depth, terminological precision, and interpretive caution expected of research dealing with adolescent mental health. Clarifying key constructs, justifying the choice of instruments, and offering a more nuanced discussion of the findings are essential steps before the manuscript can be considered for publication.
The text has been completely revised based on your suggestions.

Reviewer 3 Report
Comments and Suggestions for Authors
The article discusses the relationship between chewing speed and the occurrence of mental health problems in adolescents and young adults at a university, determining the association between risk questionnaires for mental health problems and physical activity with chewing speed when chewing gum.
The sample has some biases that warrant comment. There is no description of how the sample was recruited. Advertisements? From hospital outpatient clinics? The mix of students aged 12 to 24 implies that they are adolescents and young adults, whereas the World Health Organization defines adolescence as up to 19 years of age and young adulthood as up to 25 years of age. The nutritional analysis of adolescents is typically evaluated using the criteria of the same organization, plotting BMI and height-for-age indices on WHO charts (the same as those used by the US NCHS), which have been adapted and adjusted to pediatric charts for ages 0 to 5 years. BMI indices in adolescents differ from those in adults. The Rohrer index, which measures proportionality, is widely used in newborns.
The lack of groups from other social and economic levels prevents the results from being extrapolated.
Numerous methods for assessing chewing have been reported in the literature, including bite force, chewing and swallowing speed, and points analyzed in sequential radiographs, among others. Most studies indicate that faster chewing speeds are associated with weight gain, worsening nutritional status, and an increased risk of being overweight or obese. Hypothesis is probably due to a lower perception of taste and a greater need for portions to taste the food. Anxiety, compulsion, and lack of control have also been associated with this. However, we must consider that a single type of stimulus alone cannot determine chewing speed, as gum may be unfamiliar to some participants. In contrast, others may already be accustomed to chewing gum.
On the other hand, considering the fact that scoring on a general questionnaire of habits as a mental disorder is highly inappropriate, as there are other psychometric forms of assessment of problems, such as Beck's for depression, and others for anxiety, motivation, self-esteem, body shape, and others. However, psychometrics does not replace clinical examination, medical history, and diagnosis by specialists.
The GHQ questionnaire is quite broad and only shows current risk behaviors. It may be related to social, economic, relationship, and self-image problems. However, it does not diagnose mental issues, but rather risk.
The results show only association and risk, not causality, and may lead to errors. A screening questionnaire, whether behavioral or chewing-related, does not allow us to say that the participant has a mental disorder or chewing problems. The conclusion must obviously be modified.
Despite the well-written text, the tables are confusing and cross-reference all study variables, with a confusing dichotomous analysis. Is the primary variable only chewing speed? Extrapolating that if the participant responds that they are more or less sleepy, this determines their chewing status, suggests that this needs to be better evaluated. If the participant is just more tired and chews more slowly, or is preoccupied with something external, it does not allow for the determination of mental risk.
The analysis of the discussion combines articles from various aspects, interconnecting everything with everything. It should be reevaluated, focusing solely on establishing behavioral risk rather than causation and effect.
Author Response
The comments have been helpful for us to revise our manuscript. We have therefore thoroughly revised the manuscript on the basis of your suggestions. The parts of the manuscript that have been significantly revised are shown in red font.
The sample has some biases that warrant comment. There is no description of how the sample was recruited. Advertisements? From hospital outpatient clinics? The mix of students aged 12 to 24 implies that they are adolescents and young adults, whereas the World Health Organization defines adolescence as up to 19 years of age and young adulthood as up to 25 years of age. The nutritional analysis of adolescents is typically evaluated using the criteria of the same organization, plotting BMI and height-for-age indices on WHO charts (the same as those used by the US NCHS), which have been adapted and adjusted to pediatric charts for ages 0 to 5 years. BMI indices in adolescents differ from those in adults. The Rohrer index, which measures proportionality, is widely used in newborns.
In accordance with your suggestions, we used the WHO chart to determine the obesity level of subjects aged 12 to 19 years. We reran the statistics and the revised results are shown in red.
The lack of groups from other social and economic levels prevents the results from being extrapolated.
We added the description of the lack of groups from other social and economic levels in the study limitations section.
Numerous methods for assessing chewing have been reported in the literature, including bite force, chewing and swallowing speed, and points analyzed in sequential radiographs, among others. Most studies indicate that faster chewing speeds are associated with weight gain, worsening nutritional status, and an increased risk of being overweight or obese. Hypothesis is probably due to a lower perception of taste and a greater need for portions to taste the food. Anxiety, compulsion, and lack of control have also been associated with this. However, we must consider that a single type of stimulus alone cannot determine chewing speed, as gum may be unfamiliar to some participants. In contrast, others may already be accustomed to chewing gum.
Based on our results and your suggestion, we added the following sentence to the Discussion section: “However, bivariate analysis showed that the BMI was weakly negatively correlated with the number of chews and the chewing time before swallowing. These results suggest BMI may be indirectly related to speed eating. It has been reported that obese children have lower taste sensitivity than non-obese children, indicating that they re-quire a greater amount of food to taste it [43]. In this study, gummy jelly was used as a test food to assess speed eating behavior. We need to consider that chewing speed cannot be determined by a single stimulus alone, as some participants may not have been accustomed to chewing gummy jelly, while others may have been.”
On the other hand, considering the fact that scoring on a general questionnaire of habits as a mental disorder is highly inappropriate, as there are other psychometric forms of assessment of problems, such as Beck's for depression, and others for anxiety, motivation, self-esteem, body shape, and others. However, psychometrics does not replace clinical examination, medical history, and diagnosis by specialists.
In accordance with your suggestions, we added the description of the Interpretation of the mental health categorization in this study in Methods section. Additionally, we have changed our interpretation of the GHG-12 results.
The GHQ questionnaire is quite broad and only shows current risk behaviors. It may be related to social, economic, relationship, and self-image problems. However, it does not diagnose mental issues, but rather risk.
In accordance with your suggestions, we added the description of the Interpretation of the mental health categorization in this study in Methods section. Additionally, we have changed our interpretation of the GHG-12 results.
The results show only association and risk, not causality, and may lead to errors. A screening questionnaire, whether behavioral or chewing-related, does not allow us to say that the participant has a mental disorder or chewing problems. The conclusion must obviously be modified.
In accordance with your suggestions, we modified the conclusion.
Despite the well-written text, the tables are confusing and cross-reference all study variables, with a confusing dichotomous analysis. Is the primary variable only chewing speed? Extrapolating that if the participant responds that they are more or less sleepy, this determines their chewing status, suggests that this needs to be better evaluated. If the participant is just more tired and chews more slowly, or is preoccupied with something external, it does not allow for the determination of mental risk.
The primary variable in this study was the swallowing threshold, and the number of chews and chewing time before swallowing, which are significantly correlated with the swallowing threshold, were also taken into consideration. We have added information about the characteristics of the swallowing threshold to the Introduction section. In accordance with your suggestions, we have revised the discussion regarding the relationship between poor sleep quality and speed eating.
The analysis of the discussion combines articles from various aspects, interconnecting everything with everything. It should be reevaluated, focusing solely on establishing behavioral risk rather than causation and effect.
In accordance with your suggestions, we have revised the text in Discussion section.

Round 2
Reviewer 1 Report
Comments and Suggestions for Authors
The manuscript has been improved following the instructions received, so it could be published in its current form.
Author Response
The comments have been helpful for us to revise our manuscript. We have therefore thoroughly revised the manuscript on the basis of your suggestions. The parts of the manuscript that have been significantly revised are shown in blue font.
The manuscript has been improved following the instructions received, so it could be published in its current form.
Thank you for your generous comment.
Reviewer 2 Report
Comments and Suggestions for Authors
The authors missed some important aspects, noted previously in my review. They are all in my previous comments. Among them:
- swallowing threshold is still not defined
- the confusion regarding mental health disorders persits. The worst paragraph that puts together mental health and unhappiness is still present (see bellow) but there are many places where the authors confirm that they are not aware about what it means "mental health" .
>
> . "Mental health disorders, 28 such as depression, unhappiness, and anxiety, may lead to under- or over-eating [6,7]. 29 Another study reported that individuals with high levels of emotional eating and uncon- 30 trolled eating behaviors are at a higher risk of mental health deterioration [8]. Thus, 31 changes in mental health during adolescence may contribute to or hinder healthy eating 32 habits, and may even perpetuate poor eating habits and abnormal eating behaviors in the 33 future".
Author Response
The comments have been helpful for us to revise our manuscript. We have therefore thoroughly revised the manuscript on the basis of your suggestions. The parts of the manuscript that have been significantly revised are shown in blue font.
The authors missed some important aspects, noted previously in my review. They are all in my previous comments. Among them:
- swallowing threshold is still not defined
We added the description of the definition of swallowing threshold in Introduction section.
- the confusion regarding mental health disorders persits. The worst paragraph that puts together mental health and unhappiness is still present (see bellow) but there are many places where the authors confirm that they are not aware about what it means "mental health".
> . "Mental health disorders, such as depression, unhappiness, and anxiety, may lead to under- or over-eating [6,7]. Another study reported that individuals with high levels of emotional eating and uncontrolled eating behaviors are at a higher risk of mental health deterioration [8]. Thus, changes in mental health during adolescence may contribute to or hinder healthy eating habits, and may even perpetuate poor eating habits and abnormal eating behaviors in the future".
Thank you for your meaningful comment. In accordance with your suggestions, we corrected the text.
Reviewer 3 Report
Comments and Suggestions for Authors I still find the results presented in the tables extremely difficult to understand. The discussion should emphasize that the risk factors are mainly the relationship between nutritional status and chewing, but chewing and swallowing are important issues in determining obesity. The question is whether the methods we have are sufficient to establish this fact.
Author Response
The comments have been helpful for us to revise our manuscript. We have therefore thoroughly revised the manuscript on the basis of your suggestions. The parts of the manuscript that have been significantly revised are shown in blue font.
I still find the results presented in the tables extremely difficult to understand. The discussion should emphasize that the risk factors are mainly the relationship between nutritional status and chewing, but chewing and swallowing are important issues in determining obesity. The question is whether the methods we have are sufficient to establish this fact.
Thank you for your meaningful comment.
In Discussion section, we added the following description:
In this study, we used same type of gummy jelly as the test food to assess 20-second masticatory performance and swallowing threshold. Our results showed that the 20-second masticatory performance and maximum occlusal force in the low swallowing threshold group were significantly lower than the normal swallowing threshold group. These results suggest that participants with low swallowing thresholds may have weaker occlusal force, fewer chewing cycles, or both. Therefore, we believe that our swallowing threshold assessment is useful and reliable.